# Deep Learning-Based Classification and Semantic Segmentation of Lung Tuberculosis Lesions in Chest X-ray Images

**DOI:** 10.3390/diagnostics14090952

**Published:** 2024-04-30

**Authors:** Chih-Ying Ou, I-Yen Chen, Hsuan-Ting Chang, Chuan-Yi Wei, Dian-Yu Li, Yen-Kai Chen, Chuan-Yu Chang

**Affiliations:** 1Division of Chest Medicine, Department of Internal Medicine, National Cheng Kung University Hospital, Douliu Branch, College of Medicine, National Cheng Kung University, Douliu City 64043, Taiwan; budda0714@gmail.com (C.-Y.O.); dnrayen@hotmail.com (I.-Y.C.); 2Photonics and Information Laboratory, Department of Electrical Engineering, National Yunlin University of Science and Technology, Douliu City 64002, Taiwan; m10912025@gemail.yuntech.edu.tw (C.-Y.W.); m11012031@yuntech.edu.tw (D.-Y.L.); m11112037@yuntech.edu.tw (Y.-K.C.); 3Department of Computer Science and Information Engineering, National Yunlin University of Science and Technology, Douliu City 64002, Taiwan; chuanyu@yuntech.edu.tw

**Keywords:** chest X-ray, tuberculosis lesion, artificial intelligence, deep learning, U-Net, semantic segmentation, ensemble classifier

## Abstract

We present a deep learning (DL) network-based approach for detecting and semantically segmenting two specific types of tuberculosis (TB) lesions in chest X-ray (CXR) images. In the proposed method, we use a basic U-Net model and its enhanced versions to detect, classify, and segment TB lesions in CXR images. The model architectures used in this study are U-Net, Attention U-Net, U-Net++, Attention U-Net++, and pyramid spatial pooling (PSP) Attention U-Net++, which are optimized and compared based on the test results of each model to find the best parameters. Finally, we use four ensemble approaches which combine the top five models to further improve lesion classification and segmentation results. In the training stage, we use data augmentation and preprocessing methods to increase the number and strength of lesion features in CXR images, respectively. Our dataset consists of 110 training, 14 validation, and 98 test images. The experimental results show that the proposed ensemble model achieves a maximum mean intersection-over-union (*MIoU*) of 0.70, a mean precision rate of 0.88, a mean recall rate of 0.75, a mean F1-score of 0.81, and an accuracy of 1.0, which are all better than those of only using a single-network model. The proposed method can be used by clinicians as a diagnostic tool assisting in the examination of TB lesions in CXR images.

## 1. Introduction

According to the World Health Organization’s (WHO) Global Tuberculosis Report of 2021, tuberculosis (TB) is the 13th leading cause of death worldwide, affecting an estimated 9.9 million people per year. The report states that every year, 1.3 million people die from this disease, and 98% of them in low- and middle-income countries [1]. TB is caused by Mycobacterium tuberculosis infection, which most often affects the lungs (this is called pulmonary TB) and is spread by airborne transmission. If this disease is detected early, it can be treated with a course of antibiotics for six months, and the spread of the disease can be limited [2].

Chest X-ray (CXR) is an affordable and rapid diagnostic technique for detecting pulmonary TB [3]. There are several types of lesions associated with pulmonary TB that can be identified from a CXR image, such as nodular and cavitary lesions, pleural effusions, infiltrations/bronchiectasis, and opacities/consolidations. Nodular lesions are small, roundish spots that can appear in the lungs and may be associated with TB. Cavitary lesions are larger, hollow areas that may indicate a more active form of the disease. Pleural effusion, the most common form of extrapulmonary TB, is characterized by an intense chronic accumulation of fluid and inflammatory cells in the pleural space. Infiltrations/bronchiectasis are areas of increased bronchovascular markings with ill-defined margins that can indicate a local inflammatory response within the bronchioles. Finally, opacities/consolidations are mass-like lesions that may manifest as multi-focal involvement in the lungs. Depending on the types and locations of lesions which are indicative of pulmonary TB in CXR images, physicians may need to further investigate patients with additional tests, including sputum collection or surgical biopsy, for a definite diagnosis of TB.

In addition to diagnoses conducted by radiologists, computer-aided diagnosis using digitized CXR images has been shown to provide significant contributions to TB lesion detection. Automatic TB screening and constructing datasets of annotated CXR images have been significant research findings in TB-related topics [4,5,6,7,8,9]. Recently, the deep learning technique has received great attention due to its ability to perform image classification and organ/lesion segmentation for TB in medical image processing [10,11,12,13,14,15,16,17,18,19,20,21,22,23,24,25,26,27]. By collecting a large enough dataset, researchers can design a neural-network-based model using a supervised learning/training process. Significant features in the training data can be automatically extracted and used to build an efficient network model. For example, in a study investigating common thorax diseases, the researchers trained their deep learning networks with a large number of images and annotations using the public CXR dataset ChestX-ray8 [28]. In addition, a benchmark called ChestX-Det10 [29] is proposed to provide bounding-box annotations of ten categories of thoracic diseases or abnormalities from a subset of the ChestX-ray14 dataset. This shows the feasibility of a computer-aided diagnosis based on deep learning, with promising results for clinicians.

Among the various network architectures used in deep learning techniques, the U-Net has shown its significance for and great performance in medical image segmentation [30,31,32]. In addition, accumulating studies using the U-Net model have promising results in lung segmentation in CXR images [33,34] and further studies have already been conducted in TB-specific images [12,21,35]. However, when reviewing these published studies, which use various datasets [29,36,37,38], the precise size and exact location of TB lesions annotated using a closed contour are not provided, and all of these studies lack object-level annotations. Furthermore, there is no detailed description of TB-consistent lesions, which contain several types of radiologic findings. Additionally, these studies only contain a limited number of images and selecting appropriate data augmentation schemes is difficult. Therefore, their segmentation results need to be improved.

In this study, we created our own CXR dataset, specifically for infiltrations/bronchiectasis and opacity/consolidation lesions, which are the two most common presentations of TB in CXR images. We collected CXR images from National Cheng Kung University Hospital, Douliu Branch, and the exact locations of both lesion types in the CXR images were manually identified by two experienced pulmonologists. We then implemented a multiclass semantic segmentation method for TB lesions based on deep learning networks, including the basic U-Net, Attention U-Net, U-Net++, Attention U-Net++, and PSP Attention U-Net++ models. Finally, an ensemble model based on a linear regression scheme was used to combine the above five networks and to further improve their performance. By using an adequate training process with the correct network parameters to avoid model overfitting, the experimental results of the test images show that the proposed method can efficiently detect and segment two-class TB lesions in CXR images.

A common labeling scheme for TB lesions in CXR images is delineating them with a rectangular bounding box [22,29,39]. In addition, heatmaps, which utilize different color distributions to represent the likelihood of TB lesions, are frequently used as well [7,14,15,19,20,21,24,25,26]. However, the shape of a real lesion is hardly rectangular, and heatmaps provide even less information about the location and shape of lesions. Only a few public datasets or studies providing semantic segmentation results for different TB lesions are available [37]. It is important to be able to obtain more detailed information about detected lesions. Therefore, in this work, we not only detect and classify but also extract the exact location and shape of the two major types of TB lesions.

The remainder of this paper is organized as follows: first, the backgrounds and relevant techniques used to obtain CXR images and perform TB lesion detection are described in Section 2. Section 3 deals with the proposed methods. The experimental results are provided in Section 4. Finally, the conclusion is drawn in Section 5. 

## 2. Backgrounds and Related Techniques

CXR is usually used as the first radiological examination and remains a core technique for the screening, diagnosis, and treatment of various pulmonary diseases [40]. There are three main types of CXR based on the position and direction of the patient in relation to the X-ray source and detector panel: posterior–anterior (PA), anterior–posterior (AP), and lateral. As shown in Figure 1, PA frontal CXR images were used in our experimental dataset.

TB lesions in the thorax mainly include five types: infiltration/bronchiectasis, opacity/consolidation, cavity, pleural effusion, and miliary nodule. Due to the data available from our collaborating hospital, in this study, we only consider the most common two types of lesion pattern: infiltrations/bronchiectasis and opacities/consolidation lesions. During our image collection process, there was no special selection in terms of the distribution of lesions. We selected all images that were available at our hospital due to the limited number of available images. Images with lesion types other than the two types considered in this study were excluded. As shown in Figure 2a, infiltrations/bronchiectasis lesions have a “tram-track” appearance, with parallel and ring-like opacities in CXR images, which is related to the thickened walls of dilated bronchi [41]. A frontal CXR image shows parallel linear opacities indicative of bronchial dilatation. Figure 2b shows an example of opacity/consolidation lesions, which are characterized by mass-like patches or large air space consolidation in CXR images.

Figure 3a shows the network architecture of U-Net, which is often used for medical image segmentation tasks as it provides effective segmentation results. The U-Net network is divided into two main parts: first, the encoder path consists of two consecutive 3 × 3 convolution layers for each sample, followed by a rectified linear unit (ReLU) and a 2 × 2 maximum pooling layer. The second part of the U-Net is called the decoder path, where each up-sample uses a 2 × 2 deconvolution of the feature map until it is the same pixel size as the input image. The U-Net architecture is capable of adding encoder paths using different backbones, which define the arrangement in the encoder path. In this study, we use either ResNet-50 or DenseNet121 as the backbone.

Many improved versions of U-Net have been proposed to achieve better segmentation performance. Therefore, we also implement four U-Net-based models, which are briefly described below. Figure 3b–e show the network architectures for the other four U-Net-based models. As shown in Figure 3b, the design of U-Net++ is based on the structure of DenseNet [42], which uses a dense skip-connected grid for the middle layer between the encoder and decoder paths. This network architecture enhances the network by propagating more semantic information between the two paths. Each skip-connected grid obtains all previous feature maps for the same level. Thus, each level is the equivalent of a dense module, which is designed in a way that significantly reduces the loss of semantic information between the two paths.

Oktay et al. proposed Attention U-Net, which is implemented through the use of attention gates [43]. Figure 3c demonstrates this network’s architecture. The main function of an attention gate is to remove irrelevant features. Each layer in the extended path has an attention gate that must be passed before connecting to the up-sampled features in the extended path. The use of repeated attention gates can significantly improve segmentation performance without adding too much computational complexity to the model.

Li et al. proposed Attention U-Net++ to improve the U-Net++ segmentation architecture by using an attention gate approach [44]. As shown in Figure 3d, each convolutional block in the encoder extracts feature information, which is then transferred to the decoder through dense skip connections. The attention gates are inserted between nested convolution blocks. It can integrate hierarchical levels of feature information to improve the segmentation result.

Zhao et al. proposed a pyramid scene parsing network (PSPNet) implemented by using a pyramid space pooling (PSP) module [45]. The PSP module performs global average pooling for different sizes of values in the feature map. The PSP module is used in four different sizes, 1 × 1, 2 × 2, 3 × 3, and 6 × 6, to generate a maximum pooling pyramid structure, which stacks the final characteristic output. Figure 3e demonstrates the network architecture. Here, the Attention U-Net++ network model is modified with the PSP module, which is added between the down-sampling and up-sampling paths to obtain different sizes of image features and collect different levels of information.

Table 1 shows the weblinks to the source codes of the network architectures shown in Figure 3, except for Attention U-Net++. In our study, the code for Attention U-Net++ was derived by combining the codes of U-Net++ and Attention U-Net++.

## 3. Methodology

### 3.1. System Architecture Overview

The system architecture used in this study can be divided into three major parts: (1) dataset partition; (2) the training stage; (3) the test stage. Figure 4 shows the dataset partition scheme. First, we transformed the original CXR images from the DICOM (Digital Imaging and Communications in Medicine) format into 8-bit grayscale images in the PNG (Portable Network Graphics) format. Next, we separated the collected CXR images into a training set, a validation set, and a test set. We used the same preprocessing methods for each dataset and data augmentation for the training set. In the training stage, the training set and the validation set were fed into a deep learning network to train the model. After obtaining the trained model, the test images were sent to the model to detect the lesion type and obtain the segmentation results.

Figure 5a shows the detailed steps of the network model training stage. The training inputs include three parts: (1) preprocessed grayscale CXR images for training, as shown in Figure 3; (2) corresponding lesion regions labeled by a physician; (3) selected network model and training parameters. Figure 5b shows a block diagram of the proposed method in the test stage. First, an input grayscale CXR image is processed with the same preprocessing scheme used in the training stage. Then, the image is sent to the trained network model for lesion detection and segmentation. In the postprocessing step, a morphological opening operation is applied onto the result of detected lesion region segmentation. This tiny region, which is less than the structural element in the opening operation and is considered noise, can be removed. Then, the detected lesion type and region are added onto the original CXR image for better observation in the resultant image.

### 3.2. Image Dataset Collection

Our dataset currently consists of 222 CXR images, which include 142 images with TB lesions and 80 normal images, and all were collected from National Cheng Kung University Hospital, Yunlin Branch. Two CXR experts annotated the reference images with the details shown in Table 2. They identified the major types of lesion pattern in TB, i.e., infiltration/bronchiectasis and opacity/consolidation, which are defined as bronchoalveolar spreading (tree in bud) and small nodular patches or large air space consolidations, respectively. Figure 6a–d show selected examples CXR images from our dataset with and without the mentioned lesion patterns. Figure 6a shows a normal CXR image without any lesions. Figure 6b shows an example of an opacity/consolidation lesion, shown with a red contour in the CXR image. Figure 6c shows an example of an infiltration/bronchiectasis lesion, shown with a yellow contour in the CXR image. Figure 6d shows a CXR image with both of these TB lesions. As shown in Figure 4, the datasets are divided into three parts, of which 49.6% are for training, 6.3% are for validation, and 44.1% are for the final testing. In order to enhance the network’s learning capability, the training images were augmented eight times using the operations of random rotations, panning, and horizontal flips. Table 3 shows the exact numbers of images in the three datasets in this study. In addition to the eighteen test images with lesions, three normal CXR images without any lesions were considered as well.

### 3.3. Image Preprocessing and Data Augmentation

Selvan et al. proposed an efficient lung segmentation method for abnormal CXR images [46]. This method uses the U-Net segmentation model with variational data imputation, which facilitates the segmentation of lung lesions. By applying this method to a CXR image, we can obtain a mask of the lung region according to the lung segmentation result. Rajaraman et al. proposed a ResNet-based bone suppression model [47] in which the convolutional block has 64 filters of size 3 × 3 and which uses zero padding to preserve input size. In total, 16 ResNet blocks are used in the proposed model. This model identifies and removes bony structures from frontal CXR images to assist in reducing possible errors in interpreting TB lesions for both radiologists and deep learning models. The texture features of TB lesions in the original CXR images are usually not obvious. Therefore, they may not be enough for direct detection and diagnosis using deep learning-based methods. In this study, CXR images were preprocessed by applying image enhancement methods [48], such as the CLAHE method [49], to enhance the details, textures, and low-contrast regions of the images. 

There are eight possible combinations of the three preprocessing schemes. In our experiments, these eight combinations, listed in Table 4, were used. We compared the detection and segmentation results achieved by the various U-Net models based on these eight combinational schemes. Figure 7a–e show example images in the corresponding preprocessing steps. The original CXR image is shown in Figure 7a. As shown in Figure 7b, a 1:1 extraction of the ROI position was performed using the mask of the lung segmentation network. The CXR image of the ROI extraction result was resized into 512 × 512 pixels, as shown in Figure 7c. As shown in Figure 7d, the ROI extraction results were further segmented for exact lung location. This method ensures that the image of the lung region is maximized and free from deformation. Figure 7e shows the enhanced results, produced by applying the CLAHE method to the original image shown in Figure 7d. As shown in Figure 7e, more detailed structure, texture, and edge information has been successfully extracted.

The original sizes of the CXR images in the dataset vary. In the training stage, the training images were of the same size. Therefore, the CXR images were resized into 512 × 512 × 3 because their original format was color (red, green, and blue channels). Data augmentation schemes, which include random rotation by 10 degrees, translation, and horizontal flip [50], were used to increase the number of images in the training set. Rotating images helps the model learn to recognize the lesions from different viewpoints. The model becomes more invariant to rotation variations in CXR images. Shifting an image horizontally or vertically simulates changes in lesion position within the frame. The model can learn spatial invariance, making it less sensitive to lesion placement variations. Horizontal flip creates a new example with the same content but reversed. The model can enhance its ability to recognize lesions regardless of their orientation. Figure 8a–d show the results of applying various data augmentation schemes. The original input CXR image is shown in Figure 8a. Figure 8b shows the horizontally flipped CXR image. Figure 8c shows the original CXR image randomly rotated by 10 degrees. Finally, Figure 8d shows the CXR image which has been both vertically and horizontally translated from its original version in Figure 8a.

### 3.4. U-Net-Based Lesion Detection and Semantic Segmentation

Using the same training and test datasets, the five U-Net-based models mentioned in Section 2 were trained and then tested. Two kinds of backbone networks, ResNet50 and DenseNet121, were utilized in the U-Net-based models. To speed up the training process, we utilized transfer learning to initialize the weighting coefficients in the networks. During pretraining, the network is first trained by using large datasets for different tasks, and the trained weights are used for network initiation and for fine-tuning [51]. Our study used pre-training weights for ImageNet datasets in transfer learning. During the training of each network model, we considered trying different combinations of the preprocessing schemes and modifying different network parameters, including batch size, learning rate, epoch number, and various loss functions, so that the best model could be obtained.

The network output contains the lesion classification and segmentation results for a given test image. The output segmentation map may contain some tiny regions, which are obvious noise and should be filtered using the correct image filtering scheme. Therefore, we applied morphological opening to the lesion regions in the segmentation map. A square structural element (SE) of size 25 × 25 was selected according to the minimal region that can be considered a lesion. Next, the detected lesion was pasted onto the original CXR image and then the output image was converted back to its original size. Figure 9 shows an example of applying this postprocessing scheme on the network output. As shown in Figure 9a, two different types of TB lesion are detected. The yellow and red regions denote opacity/consolidation and infiltration/bronchiectasis lesions, respectively. However, the red region is smaller than the SE, so it is discarded using the morphological opening operation. Figure 9b shows the second step in postprocessing. The output segmentation map is resized to be equal to the lung ROI and then overlaid onto the original CXR image. Therefore, we can observe the detected lesion region and recognize the lesion type by its color in the resultant image. Note that the ground truth, denoted with a yellow contour, is also displayed for comparison.

### 3.5. Ensemble Methods

In addition to separately utilizing U-Net-based models for the semantic segmentation of TB lesions, we also constructed a combinational ensemble model. These models have shown promising results in image segmentation frameworks. For example, Shakeel et al. proposed the use of ensemble classes to detect non-small-cell lung cancer in CT images [52], and Sharma et al. proposed the use of ensemble methods to detect cancerous or non-cancerous polyps in colonoscopy [53]. Rajaraman et. al. utilized several ensemble methods, which also included U-Net-based networks, to segment TB lesions [12]. Compared to the results obtained using only a single network, ensemble methods have shown obvious advantages in terms of system performance.

As shown in Figure 10, we combined the top five best-performing models, M_1_–M_5_, to construct four ensemble methods (AND, OR, logistic regression, and dense layer stacking) for the same classification and segmentation tasks. Both the AND and OR ensemble methods performed the bitwise interception and union operations, respectively, on the predicted masks of the top five models. Obviously, the AND ensemble method would reduce false positives, but produce more missing values in segmentation results, while OR would have more false positives, but fewer missing values. 

Figure 11 shows an ensemble method based on logistic regression with a learning rule for weighting adjustment. The output segmentation maps of the five models are weighted and then summed as the input of an activation function, which is a nonlinear sigmoid function. The output of the activation function is compared with the desired output. We calculated the MSE as their difference and applied the back-propagation learning algorithm to adjusting the weighting factors *w*_1_, *w*_2_, *w*_3_, *w*_4_, and *w*_5_ for the five models. This ensemble model was trained using the same training set images as the single-network model. The training process stopped when the MSE was below a threshold value or converges. Then, the weighting factors *w*_1_, *w*_2_, *w*_3_, *w*_4_, and *w*_5_ were determined and fixed. In the test stage, the final semantic segmentation result of a test image was the weighted combination of the five single-network models.

Figure 12 shows a block diagram of a stacking ensemble method based on a dense layer architecture. We constructed an integrated model whose architecture is a combination of fully connected layers with dropout functions to overcome the overfitting problem. The input data were the probabilities of concatenating the top five models to predict three classes for each pixel in the image. The number shown in the dropout function denotes the rate of randomly dropped neurons during the training stage. In the proposed stacking ensemble, three dropouts and the corresponding dense layers of various numbers of neurons with ReLU activation functions were utilized. Finally, a dense layer of three neurons with the Softmax activation function was used to determine the final result.

## 4. Experimental Results and Discussion

The experiments were performed using a Python 3.6 programming platform and a Tensorflow 2.4 framework in a PC with a 64-bit Windows 10 operating system. The major hardware specifications of the PC are Intel Core i7-10700F@2.90 GHz and GeForce RTX 3090 GPU with 24 GB DRAM. We collected 222 CXR images, all from NCKU Hospital, of which 80 are of healthy patients and the other 142 contain lesions labeled by experienced physicians. Among all the images, 110 and 14 were used for training and validation, respectively, while the other 98 (18 with lesions and 80 healthy) images were used for testing. In our study, the semantic segmentation results produced by the trained models were read by two pulmonary radiologists and compared with the ground truth. Both pulmonologists have more than 10 years of clinical experience in reading CXR images. The consistency of TB lesions between the segmentation results and the interpretations of the expert radiologists was assessed in terms of both locations and patterns in CXR images. A misalignment in segmentation between the two physicians would be referred to a radiological formal report as the final consensus.

In the training process, all five network models, U-Net, Attention U-Net, U-Net++, Attention U-Net++, and PSP Attention U-Net++, used ResNet-50 and DenseNet121 as their backbone networks, and all used transfer learning for pretraining. Using the Adam optimizer [54], we found that the best network parameters were as follows: learning rate 10^−4^, training batch size 8, and maximum epoch number 50. These hyperparameters were selected referring to the values in the literature. Regarding learning rate, we tested 10^−3^ and 10^−4^ and found that 10^−4^ can achieve better performance. Batch size 8 was used because of the limited memory size available in our PC. As for epoch number, we cannot guarantee the exact maximum number since the real epoch number depends on the convergence criterion set in the training process. In our method, the training process stops when the accuracy values of the validation data are not increased in 10 consecutive epochs. In fact, the exact epoch number was always less than 50 because the convergence criterion could be easily satisfied under 50 epochs in our experiments. The model that we saved at the epoch number with the highest accuracy values was used for the testing data. Four loss functions, namely categorical cross-entropy loss (CCL) [55], focal Tversky loss (FTL) [56], first hybrid loss (Dice Loss + FTL, DFL) [57], and second hybrid loss (IoU Loss + FTL, IFL) [58], were all tested to find the best result. CCL maximizes the likelihood of the true class given the model’s predictions. It encourages the model to assign high probabilities to the correct class. Minimizing CCL aligns with minimizing classification error. FTL is designed for imbalanced binary segmentation tasks. Standard loss functions may favor the majority class. FTL balances positive and negative samples, combines precision and recall, and emphasizing false positives and false negatives differently. On the other hand, Dice loss and IoU loss are used for image segmentation tasks. The former and latter measure the overlap and ratio of intersection to union, respectively, between the predicted and ground truth masks. Dice loss encourages precise segmentation boundaries, while IoU loss promotes better localization and segmentation. When the convergence criterion in training was satisfied, the network model weighting factors with the lowest loss for the validation set were saved and the training process was stopped. 

There were two tasks in the proposed method: (1) the detection and classification of TB lesion types; (2) the semantic segmentation of the detected lesion in the test image. To evaluate the classification performance of the proposed method, accuracy, *Acc*, defined in Equation (1), was used to represent the ratio of correctly classified cases in all test images.
(1)$$Acc=\frac{\left(TP+TN\right)}{\left(TP+FN+FP+TN\right)}$$

Here *TP*, *TN*, *FP*, and *FN* denote the numbers of true-positive, true-negative, false-positive, and false-negative cases, respectively. The evaluation functions used in the semantic segmentation results include precision rate (*P*), recall rate (*R*), F1-score (*F*1), and intersection over Union (*IoU*), and the definitions are shown in Equations (2)–(5), respectively.
(2)$$P=\frac{C\cap^{\;}G}{C}$$
(3)$$R=\frac{C\cap^{\;}G}{G}$$
(4)$$F1=\frac{2\left(C\cap^{\;}G\right)}{C+G}$$
(5)$$IoU=\frac{G\cap^{\;}C}{G\cup^{\;}C}$$

Note that *C* and *G* denote the predicted and ground truth regions, respectively. The *F*1-score is used to balance the evaluation of precision and recall rates. *IoU* is the intersection-over-union rate between the predicted result and the ground truth region. In medical diagnosis, high precision is crucial. A false positive (misclassifying a healthy patient as having a disease) can lead to unnecessary stress, additional tests, and costs. Achieving high precision ensures that the majority of positive predictions are indeed true positives. Recall is equally important. Missing a true positive (failing to detect a disease) can have severe consequences for the patient. High recall ensures that fewer cases of the lesion go undetected. The *F*1-score balances precision and recall. It is suitable for our TB lesion classification because it considers both false positives and false negatives. A high *F*1-score indicates a good trade-off between precision and recall. Finally, *IoU* is widely used in semantic segmentation. It quantifies the overlap between predicted and ground truth masks. A high *IoU* indicates accurate delineation of the detected lesion boundaries. It is especially relevant when precise localization matters.

Table 5 shows the best performance after testing the eight preprocessing schemes and the various loss functions for mean *IoU* (*MIoU*), mean *F*1 *(MF*1*)* score, *Acc*, and the number of network parameters of the five network models with two different backbones. For each network model, we found the best performance by using different preprocessing schemes while using a fixed loss function. The maximum value in each column is shown in boldface. For example, the preprocessing scheme that achieved the best performance for the U-Net model with the ResNet50 backbone using the IFL loss function was lung ROI extraction (R). As shown in Table 5, the best *MioU*, of 0.67, was obtained by using the U-Net++ model with the DenseNet121 backbone, the DFL loss function, and the RS preprocessing scheme. The best *MF*1 score, 0.79, was achieved using PSP Attention U-Net++ with the ResNet50 backone, the DFL loss function, and the RS preprocessing scheme. The best accuracy, 1.0, was obtained using Attention U-Net++ with the ResNet50 backbone, the FTL loss function, and the RB preprocessing scheme.

To further improve performance, the test results for the four different ensemble methods mentioned in Section 3 were investigated as well. According to the results shown in Table 5, we selected the top five single-network models with the corresponding backbones and preprocessing schemes. The two most important criteria for choosing the five networks were *MioU* and *Acc,* since they are the most useful in identifying the two types of TB lesions. However, U-Net models had lower *MIoU* values (0.64 and 0.62) than the other single models. According to our experimental results, we selected U-Net++ (DenseNet121, IFL, RBS) to replace U-Net because it also has a high *MIoU* value (0.66) and can lead to higher ensemble method performance than the latter. The final five single networks, with their backbones, loss functions, and preprocessing schemes, are as follows: M_1_: Attention U-Net++ (ResNet50, FTL, RB); M_2_: U-Net++ (DenseNet121, DFL, RS); M_3_: U-Net++ (DenseNet121, IFL, RBS); M_4_: Attention U-Net (DenseNet121, FTL, RC); and M_5_: PSP Attention U-Net++ (DenseNet121, CCL, R). Note that the numbers of network parameters in the training stage when using the ResNet50 backbone are more than twice as high as those achieved using DenseNet121.

Figure 13a–e show the *MIoU* and loss function curves of the five selected network models in the training process. The training processes all terminate at less than 30 epochs due to the detected convergence. 

Table 6 shows a performance comparison of the five ensemble methods in terms of *MIoU*, mean precision rate, mean recall rate, *MF*1, and *Acc*. The maximum value in each column is shown in boldface. The ensemble method using stacking dense layer achieved the best results, except for the mean precision rate. This ensemble method achieved a *MIoU* of 0.70, a mean precision rate of 0.88, a mean recall rate of 0.75, a *MF*1 of 0.81, and an *Acc* of 1.0, which are all better than the highest values achieved by all single models, as shown in Table 5. During the training stage of this ensemble method, the Adam optimizer was used. In addition, the batch size was one, the preset epoch number was 50, the learning rate was 10^−4^, and the dropout rate in the final convolutional layer was 0.5. Figure 14a,b show the *MIoU* scores and validation loss, respectively, of the training and validation curves in the training stage of this ensemble method. The epoch number stops at 37 because the convergence criterion is satisfied.

To investigate the contributions of the different five models used in the ensemble method, an ablation study was conducted. Table 7 shows a performance comparison of the original ensemble method and of the others in which one of the five single models in the original method is discarded. We can observe that the original ensemble method achieves the best *MIoU*, *MF*1, and *Acc*, while its performance could be reduced by discarding any one single model. Accuracy *Acc* is decreased if any model out of M_1_, M_4_, or M_5_ is discarded. When model M_1_ is discarded, the *MIoU*, mean recall rate, *MF*1, and *Acc* are all decreased to their corresponding minima. If we discard model M_2_, the *MIoU* is only 0.01 less than that of the original method, while the others are the same. On the other hand, the contribution of model M_3_ in the ensemble method is more significant than that of model M_2_ because the values of *MIoU*, mean precision and recall rates, and *MF*1 after discarding the model M_3_ are all lower than those obtained after discarding model M_2_. The contribution of model M_4_ is also obvious, since four of the five metrics are decreased when it is discarded. Finally, after discarding model M_5_, the mean precision and recall rates are increased but accuracy is decreased. According to the results above, we can conclude that model M_1_ has the most significant contribution to this ensemble method.

Figure 15a–d show the experimental results of the eighteen test CXR images containing at least one type of TB lesions (six test images contain both types simultaneously). Figure 15a shows the original CXR images. The ground truth masks of the two types of TB lesion are shown in Figure 15b, while the masks of detected semantic segmentation results are shown in Figure 15c. Finally, we overlaid the segmentation results in Figure 15c onto the original images in Figure 15a and obtained the postprocessed images shown in Figure 15d. Consistency is defined as compatibility of locations and patterns of TB lesions in CXR images between the segmentation results and the interpretations of expert radiologists. All of the TB lesions detected by the trained model in the 18 test CXR images meet the criteria of consistency. Two pulmonary radiologists confirmed the results by performing a visual comparison of all the test images with an acceptable *MIoU* (0.7 in our study results) between the segmentation results and ground truth data. As illustrated in Figure 15, by using the stacking ensemble method, both types of TB lesions are efficiently distinguished in all images (opacity/consolidation pattern in red and infiltrations/bronchiectasis pattern in yellow) and semantically segmented at their corresponding positions in ground truth data.

Table 8 shows a comparison between our study and three related studies, specifically in terms of TB lesion detection, classification, and visualization methods. In Refs. [12,21], TB-consistent lesions are detected and segmented without classification information. Multicategory TB lesion detection and classification are performed in Ref. [39]. However, lesions are labeled by a rectangular bounding box rather than semantical segmentation. Our approach efficiently identifies the two major types of TB lesions with their sizes and positions so that the physician can obtain more detailed information. Note that the Shenzhen TB CXR dataset is used in all the other three studies. Therefore, we will also apply our approach to this dataset so that the robustness of the proposed method can be verified.

## 5. Conclusions

This paper presents an approach to the semantic segmentation of TB lesion regions and types (currently opacity/consolidation and infiltrations/bronchiectasis) in CXR images based on U-Net-based network models. We first selected 110 images from the original CXR image dataset and used the data augmentation method to obtain 880 training images. Several image preprocessing methods were used to enhance and extract more lesion features. In the proposed method, the network models used for the semantic segmentation of TB lesions include U-Net, Attention U-Net, U-Net++, Attention U-Net++, and PSP Attention U-Net++. We experimentally investigated the performance of the various U-Net-based networks under different backbones, preprocessing schemes, loss functions, and training parameters. The experimental results showed that the best *MIoU*, *MF*1, and *Acc* are obtained using different U-Net-based network models. Moreover, the deep learning network was enhanced by applying four ensemble methods to the top five single models. Stacking ensemble using dense layer architecture for the top five models achieved the best segmentation performance. The experimental results showed that we can achieve a promising performance (1.0 *Acc*, 0.70 *MIoU*, 0.88 mean precision rate, 0.75 mean recall rate, and 0.81 *MF*1) in TB lesion classification and semantic segmentation. 

Although our approach has shown promising results, there are certain limitations. First, only two types of TB lesions were considered in this study. If more lesion types could be detected, physicians would be able to make better diagnoses. Second, changes in image quality, e.g., resolution and format, in the training stage may have affected the performance of the proposed method. Third, the dataset size and diversity, i.e., the number of the collected CXR images and patients, are not big enough considering the deep learning perspective. Therefore, the overfitting problem may exist in the current network models, and the robustness of the proposed method may be limited. Finally, classification accuracy and semantic segmentation results could be degraded when the lesion types and the number of test CXR images with lesions are increased. The utilization of more complicated deep learning networks and longer training processes will be unavoidable. 

This approach can be used to assist physicians in examining the two specific TB lesion types in CXR images. Because pulmonary TB is an infectious airway disease and is a substantial burden to public health, it is important to facilitate the detection of new cases in developing countries with a higher incidence of TB but limited medical resources. Without experienced pulmonologists or radiologists, diagnostic tools used for TB detection should be extremely well designed to allow for disease control. However, for physicians, the identification of TB images is quite challenging because the lesions are complex and comprise several subtypes. Combined with clinical symptoms indicative of TB, artificial intelligence-assisted TB detection methods can aid general physicians in cases of early suspicion of TB, allowing for immediate referral to pulmonologists without delaying diagnosis and treatment. Although not all subtypes of TB lesions were included in our dataset analysis, infiltrations/bronchiectasis and opacity/consolidation lesions are the most frequently encountered subtypes and are very difficult to detect in clinical practice. Therefore, our study results highlight the potential impact of deep learning methods as an assistant tool in the radiologic diagnosis of TB.

In our future work, we will continuously collect more CXR images for both the training and test stages from National Cheng Kung University Hospital, Douliu Branch, so that the proposed method can be more robust. By searching the public datasets available on the Internet, we may collect more useful CXR images as well, and we can verify the robustness of the proposed method on different datasets. In addition, we will extend the proposed method to be able to detect and segment more TB lesion types, such as miliary, fibrosis, and cavitation lesions, when sufficient numbers of their corresponding CXR images can be collected.

## Figures and Tables

**Figure 1 diagnostics-14-00952-f001:**
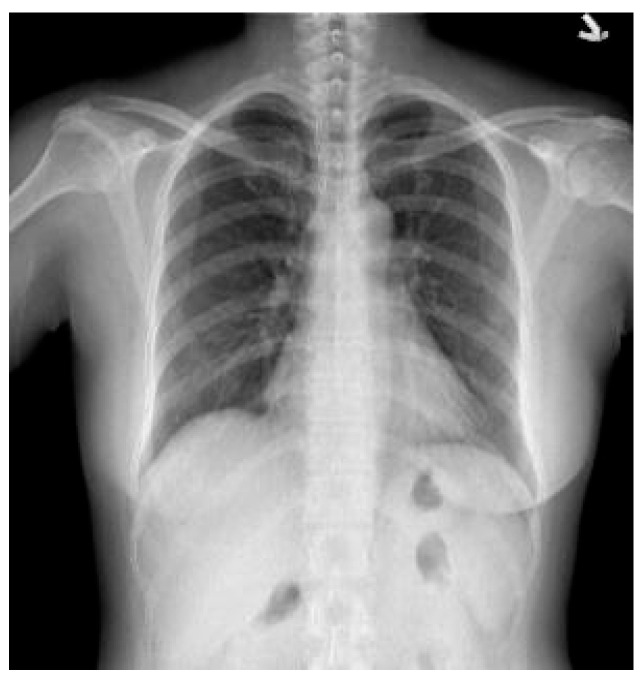
A healthy CXR image without TB lesions.

**Figure 2 diagnostics-14-00952-f002:**
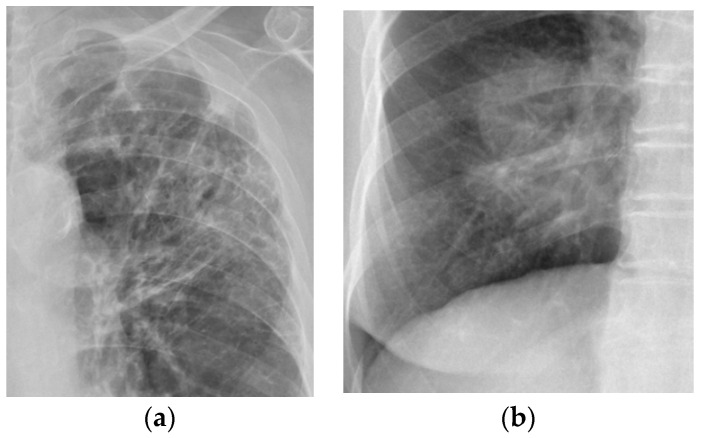
Examples of two major TB lesion types: (**a**) infiltrations/bronchiectasis; (**b**) opacity/consolidation.

**Figure 3 diagnostics-14-00952-f003:**
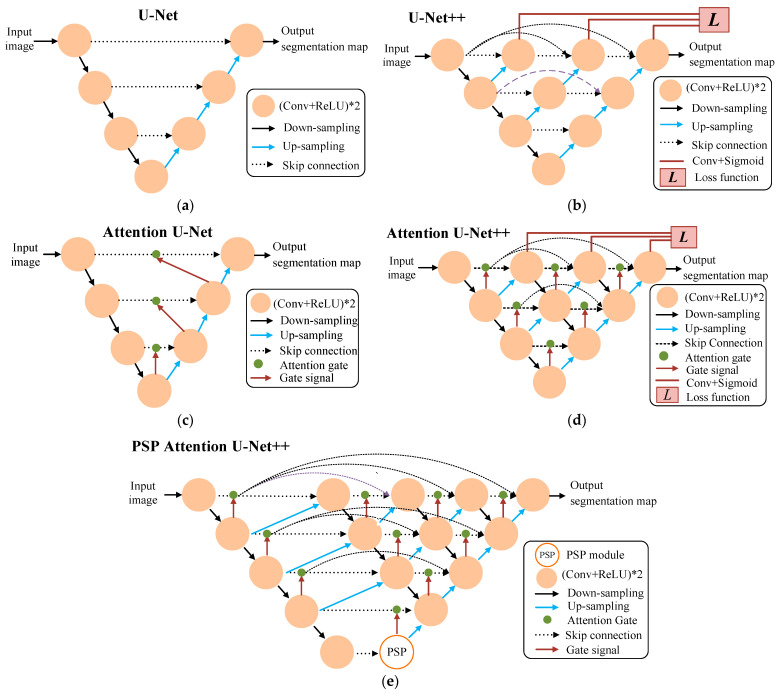
Typical network architectures of U-Net-based models used in this study: (**a**) U-Net; (**b**) U-Net++; (**c**) Attention U-Net; (**d**) Attention U-Net++; (**e**) PSP Attention U-Net++.

**Figure 4 diagnostics-14-00952-f004:**
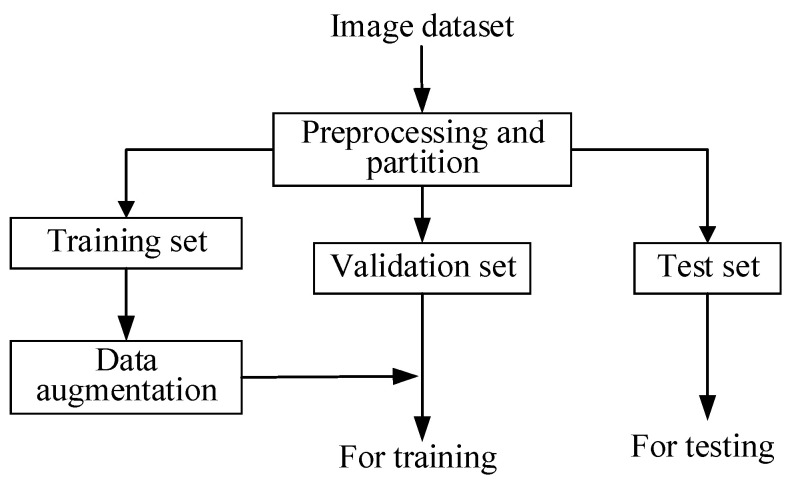
Dataset partition for training, validation, and test stages in the proposed method.

**Figure 5 diagnostics-14-00952-f005:**
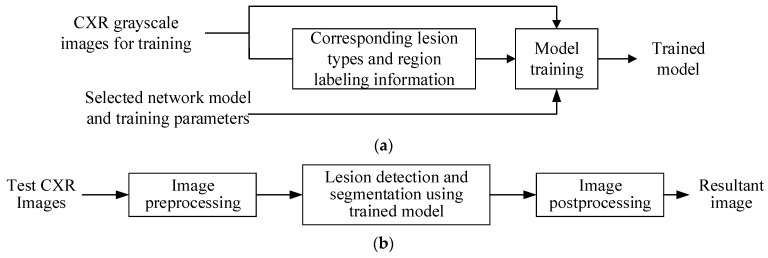
Systematic block diagrams for (**a**) training and (**b**) test stages.

**Figure 6 diagnostics-14-00952-f006:**
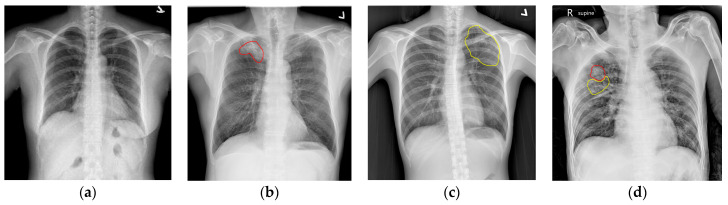
Examples of CXR images with and without TB lesions: (**a**) normal CXR image; (**b**) CXR image of opacity/consolidation lesion (marked in red); (**c**) CXR image of infiltration/bronchiectasis lesion (marked in yellow); (**d**) CXR image with both types of TB lesions.

**Figure 7 diagnostics-14-00952-f007:**
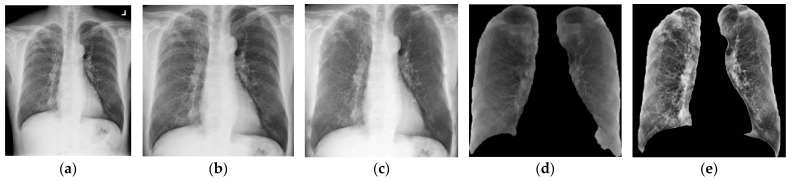
An example of CXR image enhancement by using various preprocessing schemes: (**a**) Original; (**b**) lung ROI extraction; (**c**) bone suppression; (**d**) lung segmentation; (**e**) CLAHE.

**Figure 8 diagnostics-14-00952-f008:**
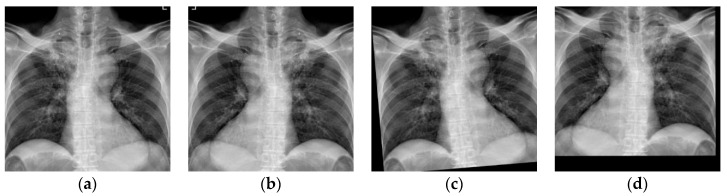
Examples of data augmentation using a CXR image: (**a**) original; (**b**) horizontally flipped; (**c**) randomly rotated by 10 degrees; (**d**) vertically and horizontally translated.

**Figure 9 diagnostics-14-00952-f009:**
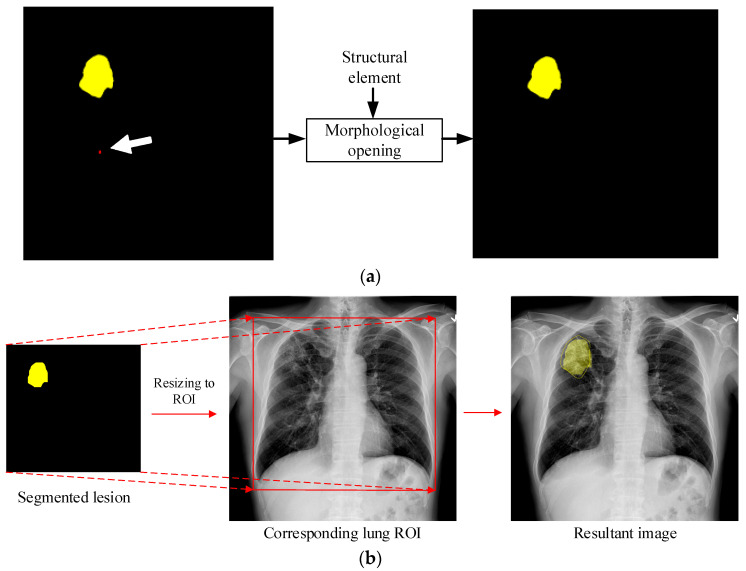
Image postprocessing schemes: (**a**) application of morphological opening to the detected segmentation map; (**b**) overlaying of the filtered lesion region onto the original CXR image.

**Figure 10 diagnostics-14-00952-f010:**
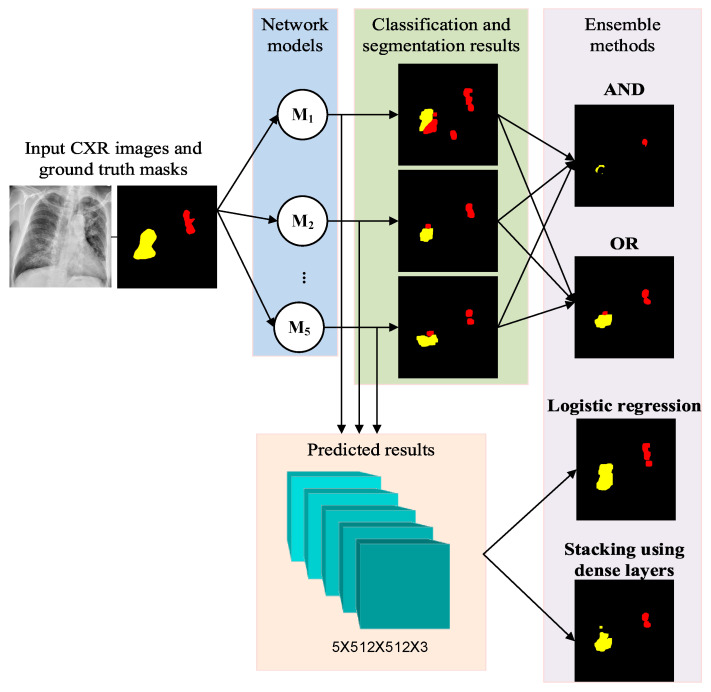
Ensemble methods based on various ensemble models.

**Figure 11 diagnostics-14-00952-f011:**
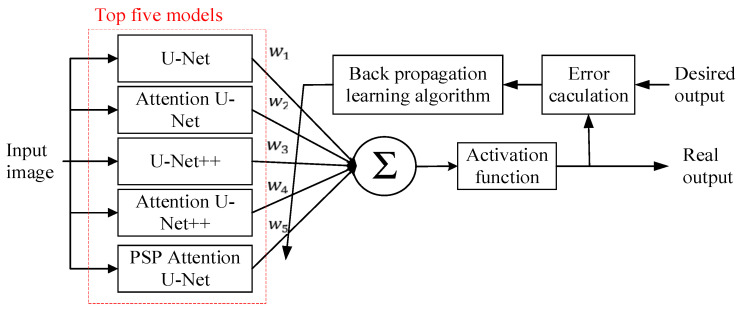
Ensemble method based on logistic regression with learning rule for weighting adjustments.

**Figure 12 diagnostics-14-00952-f012:**
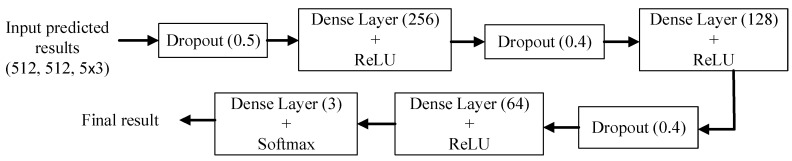
Block diagram of stacking ensemble method based on dense layer architecture.

**Figure 13 diagnostics-14-00952-f013:**
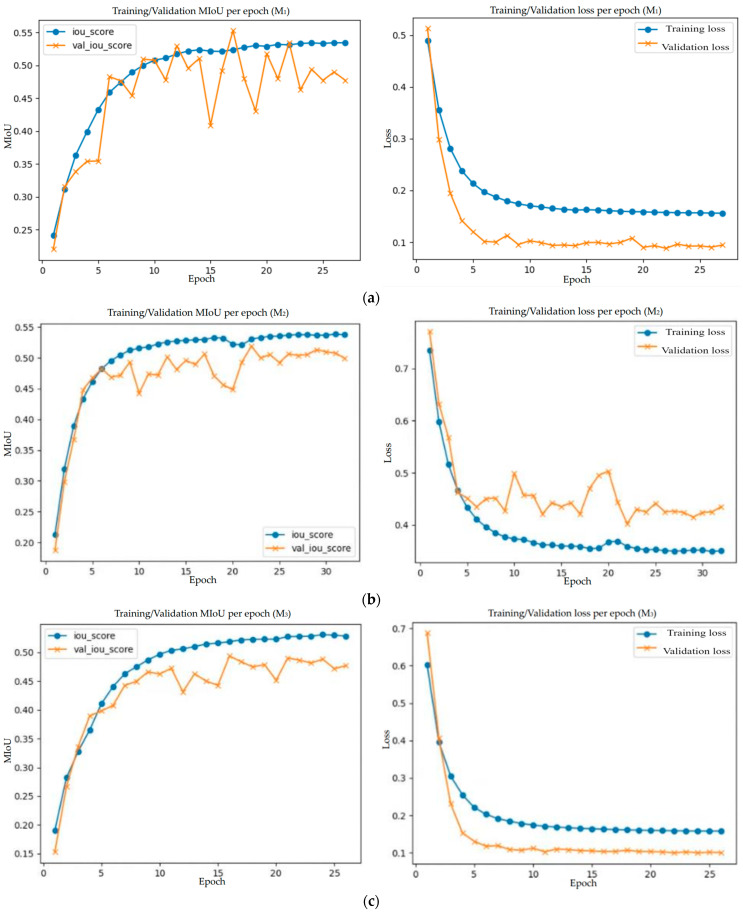
The training and validation curves of *MIoU* scores (left column) and training and validation losses (right column) achieved in the training process of the selected top five networks: (**a**) M_1_: Attention Unet++ network (ResNet50, FTL, RB); (**b**) M_2_: U-Net++ (DenseNet121, DFL, RS); (**c**) M_3_: (DenseNet121, IFL, RBS); (**d**) M_4_ (DenseNet121, FTL, RC); (**e**) M_5_ (DenseNet121, CCL, R).

**Figure 14 diagnostics-14-00952-f014:**
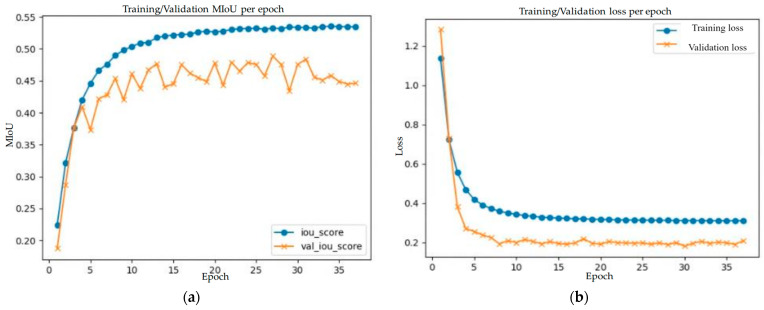
The training and validation curves during the training process of the best ensemble method using stacking dense layers (batch size 1, Adam optimizer, learning rate 0.0001, dropout rate 0.5, and epoch number 37): (**a**) *MIoU* score; (**b**) training and validation loss.

**Figure 15 diagnostics-14-00952-f015:**
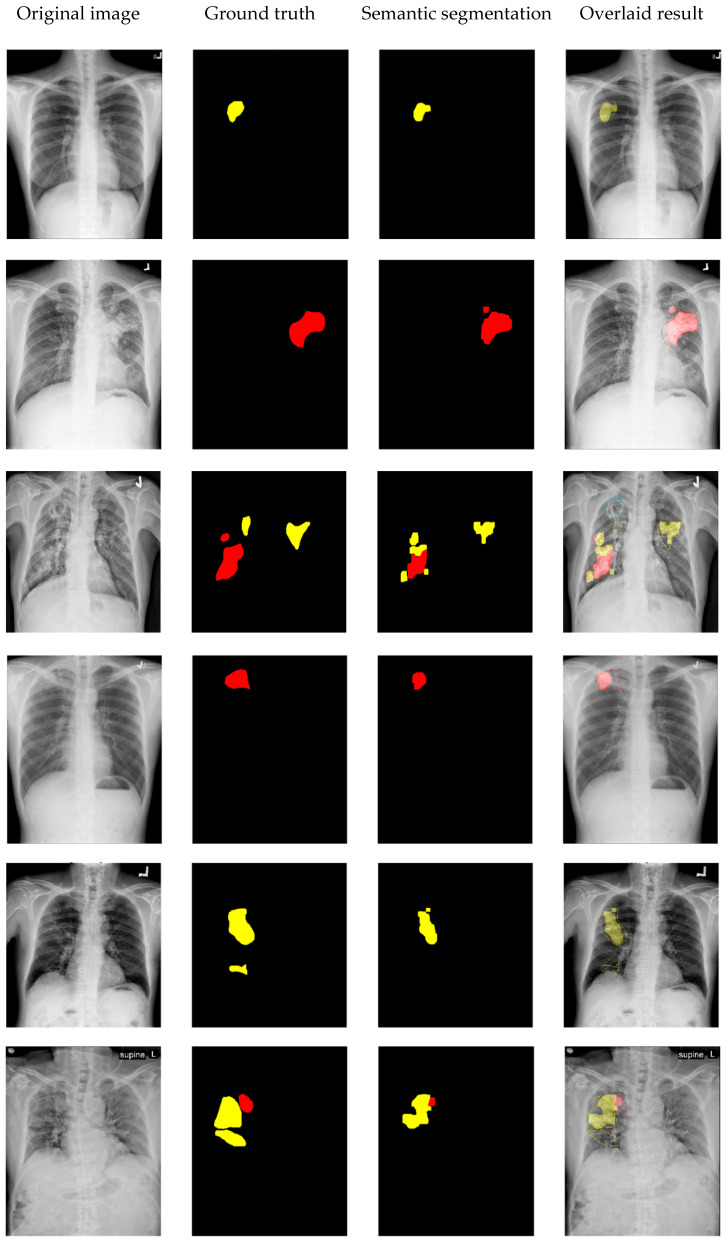
Experimental results of 18 test images: (**a**) original CXR images; (**b**) masks of lesion ground truth (red: opacity/consolidation; yellow: infiltration/bronchiectasis); (**c**) masks of semantic segmentation results; (**d**) detected lesions overlaid on the original CXR images.

**Table 1 diagnostics-14-00952-t001:** Source codes and references of the five U-Net-based network architectures.

Network Architecture	Weblink to Source Code
U-Net	https://github.com/yingkaisha/keras-unet-collection/blob/main/keras_unet_collection/_model_unet_2d.py, accessed on 30 April 2024
U-Net++ [42]	https://github.com/MrGiovanni/UNetPlusPlus/blob/master/pytorch/nnunet/network_architecture/generic_UNetPlusPlus.py, accessed on 30 April 2024
Attention U-Net [43]	https://github.com/sfczekalski/attention_unet/blob/master/model.py, accessed on 30 April 2024
Attention U-Net++ [44]	Based on U-Net++ [42] & use Attention gate function proposed in Ref. [44]. A similar version can be found in https://github.com/kushalchordiya216/Attention-UNet-plus/blob/master/unetplus.ipynb, accessed on 30 April 2024
PSP Attention U-Net++ [45]	https://github.com/hszhao/PSPNet/tree/master/src/caffe, accessed on 30 April 2024

**Table 2 diagnostics-14-00952-t002:** Dataset information.

TB Lesion Types	Number of Images
Infiltration/Bronchiectasis	41
Opacity/Consolidation	31
Both TB lesions	70
Normal	80
Total	222

**Table 3 diagnostics-14-00952-t003:** Data augmentation.

Dataset	Number of Images
Training Set	110→880
Validation Set	14
Test Set	18 w/ + 80 w/o TB lesions

**Table 4 diagnostics-14-00952-t004:** Eight combinations of different preprocessing schemes.

Abbreviation	Included Methods
R	Lung ROI extraction
RC	Lung ROI extraction and CLAHE
RB	Lung ROI extraction and bone suppression
RBC	Lung ROI extraction, bone suppression, and CLAHE
RS	Lung ROI extraction and lung segmentation
RSC	Lung ROI extraction, lung segmentation, and CLAHE
RBS	Lung ROI extraction, bone suppression, and lung segmentation
RBSC	Lung ROI extraction, bone suppression, lung segmentation, and CLAHE

**Table 5 diagnostics-14-00952-t005:** Best performance metric comparison between various loss functions and preprocessing schemes for each model. The maximum value in each column is shown in boldface.

Model	Backbone	Loss Function	Best Preprocessing Scheme	*MIoU*	*MF*1	*Acc*	Parameter Count	Selected in Ensemble Method?
U-Net	ResNet50	IFL	R	0.64	0.75	0.99	33 M	No
Unet++	ResNet50	IFL	RC	0.65	077	0.99	36 M	No
Attention U-Net	ResNet50	DFL	RC	0.66	0.78	0.98	38 M	No
Attention U-Net++	ResNet50	FTL	RB	0.64	0.78	**1.0**	43 M	M_1_
PSP Attention U-Net++	ResNet50	DFL	RS	0.65	**0.79**	0.99	48 M	No
U-Net	DenseNet121	DFL	R	0.62	0.73	0.98	12 M	No
U-Net++	DenseNet121	DFL	RS	**0.67**	0.78	0.99	14 M	M_2_
		IFL	RBS	**0.67**	0.78	0.98	14 M	M_3_
Attention U-Net	DenseNet121	FTL	RC	0.62	0.74	0.99	15 M	M_4_
Attention U-Net++	DenseNet121	DFL	RBSC	0.61	0.73	0.99	18 M	No
PSP Attention U-Net++	DenseNet121	CCL	R	0.63	0.75	0.99	21 M	M_5_

**Table 6 diagnostics-14-00952-t006:** Performance metric comparison of the four ensemble methods. The highest value for the performance metric in each column is shown in boldface.

Ensemble Method	*MIoU*	Mean Precision Rate	Mean Recall Rate	*MF*1	*Acc*
AND	0.53	**0.94**	0.54	0.64	0.99
OR	0.69	0.90	0.73	0.80	**1.0**
Logistic regression	0.69	0.85	**0.75**	0.79	**1.0**
Stacking using dense layer	**0.70**	0.88	**0.75**	**0.81**	**1.0**

**Table 7 diagnostics-14-00952-t007:** Ablation study on the top five single models (M_1_~M_5_) used in the best ensemble method with the highest *MIoU*, *MF*1, and *Acc* values. The highest value for each performance metric is shown in boldface.

Selected Models in the Ensemble Method					Performance Metrics				
M_1_	M_2_	M_3_	M_4_	M_5_	*MIoU*	Mean Precision Rate	Mean Recall Rate	*MF*1	*Acc*
√	√	√	√	√	0.70	0.88	0.75	**0.81**	**1.0**
**×**	√	√	√	√	0.68	0.88	0.73	0.79	0.98
√	**×**	√	√	√	0.69	0.88	0.75	**0.81**	**1.0**
√	√	**×**	√	√	0.68	0.87	0.74	0.79	**1.0**
√	√	√	**×**	√	0.69	0.88	0.74	0.79	0.99
√	√	√	√	**×**	**0.70**	**0.89**	**0.76**	**0.81**	0.99

**Table 8 diagnostics-14-00952-t008:** Comparisons to other similar studies on TB lesion segmentation.

Reference	Target Lesion and Classification	Lesion Visualization Method	Dataset
[12]	TB-consistent lesions w/o classification	Semantic mask	Shenzhen TB CXR
[21]	TB-consistent lesions w/o classification	GT ^1^: Rectangular bounding box Predicted: ROI mask	TBX11K CXR Shenzhen TB CXR Montgomery TB CXR
[39]	Multicategory TB lesion w/classification	Rectangular bounding box	Shenzhen TB CXR Montgomery TB CXR Local (First Affiliated Hospital of Xi’an Jiao Tong University)
Ours	Infiltration/bronchiectasis and opacity/consolidation w/classification	Semantic mask	Local (National Cheng Kung University Hospital Douliu Branch)

^1^ GT: Ground truth.

## Data Availability

Our source codes for implementing the proposed methods are available on GitHub: https://github.com/Chen-Yen-Kai/Chest and https://github.com/Chen-Yen-Kai/Chest-X-ray-Images-Using-Stacked-Ensemble-Learning.

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
