# Peer review of "Deep Learning-Based Classification and Semantic Segmentation of Lung Tuberculosis Lesions in Chest X-ray Images"

_diagnostics, 2024, doi:10.3390/diagnostics14090952_

Round 1
Reviewer 1 Report (New Reviewer)
Comments and Suggestions for Authors
Clarity and Detail in Methodology: The manuscript provides a general description of the use of U-Net-based models for lesion detection and semantic segmentation, as well as the implementation of ensemble models. However, it provides insufficient information on data augmentation techniques and their specific impact on the model's generalizability. For example, the statement "the training images are augmented eight times by using the operations of random rotations, panning, and horizontal flips" could be expanded by specifying how each augmentation technique contributes to the robustness of the model against variations in the input data.
Validation of Results: The results are promising but need to be contextualized by comparing with made with other state-of-the-art methods. For substantial validation, the authors could compare their ensemble model's performance with other leading methods in the field and highlight the relative strengths and weaknesses of the proposed method.
Dataset Description: The dataset comprises 145 CXR images with detailed annotations by experts, focusing on two types of lesions. While the data is described, other important information needed to assess model generalizability such as the characteristics of the image, distribution of the lesion, and potential biases need to be provided.
Loss Function and Optimization: The manuscript mentions the use of various loss functions like Categorical cross-entropy and Focal Tversky loss. Authors should elaborate on the rationale behind choosing these specific functions or how they influence the learning process. A detailed discussion on the suitability and impact of these loss functions on the model's performance is needed.
Hyperparameter Tuning: Hyperparameters such as learning rate, batch size, and epoch number are listed; however, the manuscript need to detail the tuning process. For instance, the range of values explored for each parameter and the criteria for selecting the optimal model configuration should be explored.
Evaluation Metrics: Standard evaluation metrics such as precision, recall, F1-scores, and Intersection over Union (IoU) are used. However, a deeper analysis of these metrics, including a discussion of their suitability for this study and the implications of the achieved scores, would strengthen the validation of the results.
Clinical Relevance: The manuscript states the potential of the proposed method as an assistant diagnostic tool, it should however include a discussion on how it might improve current diagnostic procedures for TB from a clinical perspective.
Robustness and Limitations: The robustness of the method against changes in image quality and the limitations regarding dataset size and diversity need to be addressed more thoroughly to inform the potential scope of application and the boundaries within which the proposed models are effective.
Code Availability: It should be clearly stated if the code for the models and preprocessing steps will be made available to the public, which is important for reproducibility and further research by others in the field.
Comments on the Quality of English Language
Minor English editing request
Author Response
Dear Reviewer,
Please check the attached PDF file for details, thanks.

Reviewer 2 Report (New Reviewer)
Comments and Suggestions for Authors
The paper describes a methodology for TB detection/segmentation on CXR.
5 uNET architectures are used independently and combined.
I find the approach interesting and meaningful.
There are minor things I would improve:
1. Due to their stochastic characteristic, I would find it more natural to present the networks' performance as mead+DS (in a k-fold scenario), rather than to present the best performing results.
2. Several times you state an ACC of 1, but Precision and Recall with lower values. Either the ACC refers to the Image classification task only, and P and R to the segmentation tasks, either there must be a mistake within the computation. If it is the first one, please reshape the text to make it more clear that you are reporting two tasks.
3. The dataset you are using might be of interest on the research filed, and as you are using it for the current paper, it would be nice to have it public available. The same goes for the algorithms, as you use mainstream networks. For the algorithms, the trained networks could be sufficient.
Comments on the Quality of English LanguageEnglish is clear, however some native proofreading would improve the flow of the paper.
Author Response
Dear Reviewer,
Please check the attached PDF file for details, thanks.

Reviewer 3 Report (New Reviewer)
Comments and Suggestions for Authors
Dear Authors,
The manuscript entitled “Deep Learning Based Classification and Semantic Segmentation of Lung Tuberculosis Lesions in Chest X-ray Images” is a comparative study of 5 different neural networks used to classify and segment 2 types of TB lesions in the lung using a very small dataset. The manuscript is well written and provides good experimental design. Here are my concerns:
- A large section (especially the background) is assigned to the description of known models (which might be unnecessary if referenced properly and if there were no changes made to the network architecture).
- The dataset is imbalanced in regard to normal healthy CXR images which is only included in the test set (only 3 images). Authors need to add at least 70 images of normal CXR images and look at the performance of the models. Does the normal images shows any presence of lesions or not?
- The dataset is small and as the author mentioned, there may be overfitting that is occurring training. Provide loss curves of the training. Authors should be able to either provide evidence of overfitting or include more data to the training corpus.
- The semantic segmentation results from the 6 trained models need to be read by at least pulmonary radiologists and compared with the ground truth.
- For the ground truth segmentation, how many years of experience of the radiologists have? Were there any misalignment in the segmentation between the 2 readers? If so, how were they resolved?
- What were the criteria of choosing M1 to M5 for the ensemble having these specific preprocessing and UNet baseline? The trained model M3: Unet++ (DenseNet121, RBS) cannot be found in Table 4. Please explain.
- How do the authors intend to implement this study in a clinical setup, if the segmentation can deal with only 2 types of lesions.
Thank you
Comments on the Quality of English LanguageDear Authors,
There are some minor typing errors:
1. line 79-81: what does "this work" referring to?
2. Figure 6: instead of "remarked", it should be "marked";
3. Line 339: the word "classificatgion" is incorrect.
4. Line 471: reference "39" is written in red.
5. Line 511: the word "availabe" is incorrect;
Author Response
Dear Reviewer,
Please check the attached PDF file for details, thanks.

Round 2
Reviewer 3 Report (New Reviewer)
Comments and Suggestions for Authors
Dear Authors,
The replies from the authors were not satisfactory. Data/results inconsistency still exists in the manuscript regarding the trained models, the loss functions and preprocessing scheme for each model. The authors should be transparent on the details on the trained models. If source codes were used, it should mentioned and referenced in the manuscript. If pre-trained networks were used, authors need to provide references.Authors have tested several neural networks in combination with different loss functions and preprocessing schemes. However, no clear trained model / ensemble has been selected with the best performance. Here are the comments below:
- The network architectures illustrated in Figure 3 are definitely explanatory and guide the readers. However, these neural networks are generic. If the authors are using these readily available, they should specify the source of the code of each architecture. Without any details about the hyper parameters, the number of filters, the kernel sizes, the learning rates, batch normalization, the number the epochs that each model was trained on, the chosen loss function and the preprocessing scheme for each experiment etc, these experiments are not reproducible. Authors state that they are not able to share the code, but they should provide the necessary information about the neural network implementation for a minimum of reproducibility. Authors selected 5 trained models (M1 to M5) but there is no indication in Table 4 about which experiment corresponds to which model number. More precisely, authors need to state which loss function was used for models 1 to 5.
- Authors have added 77 normal CXR images in the test set only and state that based on new experimental results, no lesions are detected in the normal CXR. However, no new experimental results were included in the manuscript. No such modifications can be seen in the quantitative analysis of the testing.
Based on my comment on imbalance dataset, normal CXR images need to be added in the training set not to the test set. Models require retraining with more normal cases in the training set. - Authors have provided the loss curves only for M1 (RestNet50, RB, FTL) = Attention U-Net ++. What about the other trained models? Note that figure caption should explicitly describe the details of the plots, for instance authors did not mention the loss function or the preprocessing scheme for that experiment. Even in the ablation study of the ensemble method, authors have not chosen which one outperforms.
- Authors mentioned that pulmonary radiologists have read the images and compared to the ground truth. Where are the scores/comments of the readers? The authors have not shown the data from the radiologists. This statement (line 391 - 392) is very confusing.
- Refer to comment 4.
- We still cannot find M3 in Table 4. Table 4 contains U-Net ++ (DenseNet121, RS) not RBS! Authors are not consistent in the results. Also in Table 4 what does the bold character mean?
- Authors have responded in a satisfactory manner.
- For figure 14, which trained model was used to generate the results?
Thank you.
Comments on the Quality of English LanguageThere are very minor errors in the language, but I would defer to the correction of the major experiments/data inconsistency.
Author Response
Please see the attachment, thanks.

Round 3
Reviewer 3 Report (New Reviewer)
Comments and Suggestions for Authors
Dear Authors,
The replies from the authors require more work and consistency. Thank you for providing further details on the source codes that were used for the experiments. Here are still concerns in the manuscript (see below):
- Based on the authors’ reply on the different network architectures used, the authors have not given any detail on what constitutes Attention U-Net++. Authors have been very brief saying that it is a combination of the UNet ++ and Attention U-Net. Was this network implemented by the authors from scratch? What are the specifics of the different layers. Authors will have to specify clearly.
Secondly, for the other networks, the source codes were provided by within these source codes there exist multiple implementations of a model. For example for the UNet ++ code provided, there are four segmentation models - including NestNet, PSPNet, UNet, and Xnet. Which one was used. It is also expected that the source codes were not modified in any capacity for the experiments described in the manuscript. If it is not the case, authors should be able to describe the changes.
Thirdly, authors have used the pyramid spatial pooling Attention UNet, which is SPP not PSP. PSP Net for Pyramid Scene Parsing Network is another network architecture - https://arxiv.org/pdf/1612.01105.pdf. Please clarify and edit the abbreviations in the manuscript. If SPP was used, then there are three types of pooling layers, which one was adopted for the model. The overall description of the methodology is still unclear and inconsistent. Authors should really pay attention on the details and scientific description of their manuscript.
- Authors mentioned that a maximum number of epoch of 100 was used. However it was usually less than 30. Authors should decide that the number of epochs was 30, and not 100 but stopped at 30.
- What do the authors mean by the following statement (line 408) “ The program cannot be successfully executed when the smaller batch sizes are chosen.”. This statement seems unnecessary. - Authors have responded in a satisfactory manner.
- Apart from the loss curve from Model M5 that is showing overfitting issue, the other 4 models trained fine. Thank you for providing the figures. However, the figure should be redone for the manuscript. The underscore punctuations are still in the axis label. All the graphs should be presented appropriately.
How did the authors choose to stop an experiment at # epoch number? - The confusion still exists. Authors state that the radiologists have looked at the machine-learning based segmentation and have compared them to the ground truth and that the segmented regions meet the criteria of consistency. What is the criteria of consistency? How was the latter evaluated? If radiologists have provided feedback, it should be clear in what way their feedback was formulated.
- Refer to comment 4.
- Please choose the abbreviation for categorical cross-entropy as either CCE or CCL, not both.
- Authors have responded in a satisfactory manner.
- Authors mentioned in the revised manuscript that the number of epoch stops at 37 because the overfitting criterion is satisfied. How was the overfitting criterion determined? If that’s the case what about M5, which clearly shows overfitting beyond epoch 15. Please provide more details.
Thank you.
Comments on the Quality of English LanguageNo comments for the moment
Author Response
Please see the attachment. Thank you!

This manuscript is a resubmission of an earlier submission. The following is a list of the peer review reports and author responses from that submission.
Round 1
Reviewer 1 Report
Comments and Suggestions for Authors
I reviewed your work titled "Deep Learning Based Classification and Semantic Segmentation of Lung Tuberculosis Lesions in Chest X-ray Images". Unet-based models were used in the paper. I don't think there is any innovation at this stage. These models were applied to a small data set created by the researchers. I do not think that the number of images used for training and testing is sufficient. Also, the absence of a Discussion section is another shortcoming. At the end of the Introduction section, the innovations of the paper and its contribution to the literature should be discussed. The results section covers only a very small part of the article. The result section needs to be expanded. Limitations of the paper should be included. I also did not understand why a dirty version of the manuscript, where the typos were corrected, was uploaded to the system. In the Discussion section, the relevant studies should be presented in a table and the innovations of this paper and its advantages over other studies should be highlighted. Was the test process done with images that were multiplexed or original images? It is important to perform the testing at the patient level to avoid memorizing the models. The step by which the work can be considered an innovation is presented in Figure 10. It is important for researchers to highlight this part. I respect your effort, but it is not possible for me to accept your article in its current form. Thank you.
Best regards.
Comments on the Quality of English LanguageMinor editing of English language required.